# Sliding Mode Control with Dynamical Correction for Time-Delay Piezoelectric Actuator Systems

**DOI:** 10.3390/ma13010132

**Published:** 2019-12-27

**Authors:** Javier Velasco, Oscar Barambones, Isidro Calvo, Joseba Zubia, Idurre Saez de Ocariz, Ander Chouza

**Affiliations:** 1Fundación Centro de Tecnologías Aeronáuticas (CTA), Juan de la Cierva 1, 01510 Miñano, Spain; idurre.saezdeocariz@ctaero.com; 2Department of System Engineering and Automation, Faculty of Engineering Vitoria-Gasteiz, University of the Basque Country (UPV/EHU), Nieves Cano 12, 01006 Vitoria-Gasteiz, Spain; oscar.barambones@ehu.eus (O.B.); isidro.calvo@ehu.eus (I.C.); ander.chouza@ehu.eus (A.C.); 3Department of Communications Engineering, Faculty of Engineering Bilbao, University of the Basque Country UPV/EHU, Plaza Ingeniero Torres Quevedo 1, E-48013 Bilbao, Spain; joseba.zubia@ehu.eus

**Keywords:** piezoelectric actuator, sliding mode control, dynamical correction, micropositioning system, time-delay mechanism, robust control

## Abstract

In piezoelectric actuators (PEAs), which suffer from inherent nonlinearities, sliding mode control (SMC) has proven to be a successful control strategy. Nonetheless, in micropositioning systems with time delay, integral proportional control (PI), and SMC, feedback control schemes have a tendency to overcompensate and, consequently, high controller gains must be rejected. This may produce a slow and inaccurate response. This paper presents a novel control strategy that deals with time-delay micropositioning systems aimed at achieving precise positioning by combining an open-loop control with a modified SMC scheme. The proposed SMC with dynamical correction (SMC-WDC) uses the dynamical system model to adapt the SMC inputs and avoid undesirable control response caused by delays. In order to develop the SMC-WDC scheme, an exhaustive analysis on the micropositioning system was first performed. Then, a mixed control strategy, combining inverse open-loop control and SMC-WDC, was developed. The performance of the presented control scheme was analyzed and compared experimentally with other control strategies (i.e., PI and SMC with saturation and hyperbolic functions) using different reference signals. It was found that the SMC-WDC strategy presents the best performance, that is, the fastest response and highest accuracy, especially against sudden changes of reference setpoints (frequencies >10 Hz). Additionally, if the setpoint reference frequencies are higher than 10 Hz, high integral gains are counterproductive (since the control response increases the delay), although if frequencies are below 1 Hz the integral control delay does not affect the system’s accuracy. The SMC-WDC proved to be an effective strategy for micropositioning systems, dealing with time delay and other uncertainties to achieve the setpoint command fast and precisely without chattering.

## 1. Introduction

Several industrial applications need subsystems able to achieve positioning with an accuracy of micrometers. Among precision micropositioning systems, piezoelectric actuators (PEAs) are highly suitable due to their excellent behavior in terms of response time, mechanical force, and extremely fine resolution [1,2,3]. The use of piezoelectric actuators (PEA) proved to be effective in diverse devices where micropositioning is required, such as valve control [4], precision stages [5,6], linear motor systems [7], and piezoelectric friction-inertia actuators (PFIAs) [8].

Nevertheless, the nonlinear nature of the piezo-driven stages causes difficulties in its use in real applications. As a consequence, characterization is a challenging problem that involves physical concepts, electrical and mechanical measurements, and numerical optimization techniques [9], especially focused on analyzing the main nonlinearities present in the piezoelectric actuators—creep, hysteresis, and dynamic behavior [1]. It is particularly difficult to deal with hysteresis, that is the dependence of the state of the system not only on the present stimulus but also on past stimuli. In PEAs, the hysteresis is presented by the relation of the electric field/polarization and the electric field/deformation. The hysteresis between the electric field and the deformation has great transcendence and may induce a severe open-loop position error at the PEA, which can reach as high as 10%–15% of the stage travel range [10].

Commonly, precision positioning systems include specific drives in order to achieve a desired movement. The control of these systems involves dealing with several mechanisms. Depending on the element distribution (masses, springs, damps, and friction forces) in the mechanism, the dynamic behavior of the system varies. A variable dynamic behavior affects the amplified PEA devices, with greater influence when larger loads are applied and time lag increases pursuant to increasing frequency [11]. Additionally, in complex devices the performance is limited by friction, which causes positioning errors [12]. These nonlinearities can cause important deviations, so that when designing a control strategy for precision positioning systems, not only has the PEA hysteresis to be taken into account, but also other nonlinearities and possible external disturbances. Several studies have been carried out to model the PEAs [1,2] and PEA mechanisms, such as PFIAs, PFIAs with amplification and multiple PFIA mechanisms [3]. A correct system model is essential to developing a successful PEA control strategy [13,14].

When controlling PEAs, the introduction of diverse models for representing the hysteresis and friction [15], such as the Preisach [16,17], Duhem, Maxwell, and Bouc–Wen (BW) [18,19], or the Prandtl–Ishlinskii models [17,20,21,22], allows minimizing positioning errors directly in open loop. Nevertheless, to correct the error caused by uncertainties and non-modeled deviations, the use of closed control loops is required. Integral proportional control (PI) [23], sliding mode control (SMC) [24], and modified SMC strategies [10,25,26,27,28] are simple closed-loop methods to correct uncertainties and errors in PEA systems, but they may become unstable or introduce an important delay between the input and the system response. 

When controlling complex mechanisms, where there is a remarkable time-delay behavior, the control strategy must deal with obsolete data or a slow response against control actions. For example, traditional feedback controls usually tend to be unstable since the errors are overcompensated. In cases where the system does not require a fast response (setpoint frequency <0.2 Hz), a sliding mode control with a time delay can improve the system control as shown in [29]; the time delay constant performs a similar function to an integral constant, taking into account previous data to reduce the permanent error. If the tracking requires higher frequencies (>10 Hz) and a fast response is required, integral gains are not indicated because they involve a slow reaction. Under those conditions, the strategies aim to adequately adjust the control signal, taking into account the system’s behavior, to achieve an effective and stable response. Researchers have been successful in using predictors that model the system performance, such as smith predictors to adequately adjust the input control and achieve a better performance [30,31]. If the system is subjected to a periodic signal reference, a sliding mode control with repetitive control can optimize the system performance, but the control must be robust enough to avoid aperiodic disturbances [32]. Other authors have proposed feed-forward controllers in combination with adaptive controllers [33] and neural network controllers [34].

Although SMC has been broadly studied in PEAs, its use has not been found for micropositioning with time delay. For this paper, we studied the development of a control strategy for a PEA positioning system which we designed. First, a micropositioning system with a 400 µm range is presented. The micropositioning PEA system is composed of several mechanisms, whose different properties affecting the control strategy are analyzed in detail in Section 2.1. Among the nonlinearities in this system, it is remarkable that the time-delay response affects the control performance. To overcome these problems, a sliding mode control with dynamical correction strategy (SMC-WDC) was developed and is presented in Section 2.2. The performance of this control scheme is evaluated and compared with other control strategies, namely PI and SMC, in Section 3. Finally, discussion conclusions are presented in Section 4.

## 2. Materials and Methods

### 2.1. PEA Positioning System

The PEA positioning system proposed in this paper aimed to control the position at the end of a cylinder tip with a precision of microns. A travel range of 200 µm and an accuracy of 5 µm were required. For this application, an amplified piezoelectric actuator was selected with an additional mechanism to provide a solid fixture and a linear guide to ensure linear displacements. The system was completed with control and monitoring equipment and a voltage amplifier. Each component is described in detail in the following sections.

#### 2.1.1. Amplified Piezoelectric Actuator

The commercial piezoelectric actuator used for the experiments was model PK2FVF1 Thorlabs (Newton, NJ, USA), an amplified piezoelectric actuator with flexure mount. The actuator consisted of a discrete piezoelectric stack mounted inside a flexure housing. The flexure mount acted as a lever arm that amplified the free-stroke displacement of the stand-alone discrete stack. The model PKFVF1 input voltage was between 0 and 75 volts and achieved a maximum displacement of 420 µm (while the non-amplified PEA version achieved 44.8 µm). On the other hand, the amplification mechanism reduced the maximum load to 100 N (400 N in the non-amplified PEA version). The main nonlinearities in this actuator were hysteresis and dynamical behavior. Because displacement is created by a piezo stack within a flexure mount, these actuators did not suffer from backlash [3].

PEA displacement was achieved between the plain base surface and the spherical contact on the top of the actuator housing. The spherical contact allows precise mounting of the load along the translation axis and prevents undesired stress along the other axes [35]. 

#### 2.1.2. Positioning Mechanism

The spherical contact in the amplified PEA was not capable of fixing additional tooling, so a positioning mechanism (PM) was added to provide a solid fixture to attach additional elements and ensure linear displacement. The PM was composed of (1) a main body, in contact with the PEA and guided by two parallel linear guides, (2) a spring holder which ensured the main body/PEA contact, and (3) a pin fixture with a standard thread to allow additional clamplings. Figure 1 shows the amplified piezoelectric actuator mounted with the position mechanism. The connection between the PEA and the positioning mechanism was ensured by the spring holder. In the case of very high accelerations in the backwards direction, an undesirable gap may appear between the parts.

The positioning mechanism incorporated additional mechanical elements into the system. In order to study the whole positioning system, the model can be divided into three elements: (1) the PEA, (2) the PEA amplifier mechanism (PAM), and (3) the positioning mechanism (PM). The mechanical model scheme is shown in Figure 2, where *x* represents the displacement, *m* is the mass, *k* is the spring constant, and *b* is the damp constant.

Previous research on the discrete-stack piezoelectric actuator showed very considerable response of this actuator [25]. To study the mechanism behavior, the displacement response in open-loop control was analyzed, where the complete positioning system was compared with the amplified PEA without additional tooling. The amplified PEA (PEA + PAM) showed a fast response, where the displacement started almost immediately after the voltage command and achieved the position in less than 0.02 s. When the amplified PEA was installed in the positioning mechanism (PEA + PAM + PM), a high time delay occurred; as shown in Figure 3, an additional 0.05 s delay was detected. Taking into account the deviations between the amplified PEA with and without position mechanism, we can conclude that the PM’s additional masses, springs, and frictions caused the delay. Consequently, when designing a successful control strategy for this application, the system modeling is a key point. 

#### 2.1.3. Voltage Amplifier

Open-loop piezoelectric controllers by Thorlabs provide precise, low-noise output voltages for the fine movement of piezoelectric actuators and stacks. Each output channel is independently controllable and provides a voltage range from 0 to 75 V. The input voltage is amplified by 7.5 volts/volt [35].

#### 2.1.4. Confocal Sensor

A Micro-Epsilon confocal chromatic measuring system (Raleigh, NC, USA) is composed of the confocal sensor model IFS2405-10 and controller IFC2421. The advantage of this sensor is that it measures distance with great precision (60 nm resolution and an accuracy of less than 2.5 μm) without coming into physical contact with the piece. Therefore, no additional mass is added to the mechanism.

#### 2.1.5. Control and Monitoring Hardware

A CompactRIO system (National Instruments, Austin, TX, USA) monitors and controls the positioning system. The control platform is composed of the following devices:cRIO-9039 CompactRIO Controller, 1.91 GHz Quad-Core CPU, 2 GB DRAM, 16 GB Storage, and Kintex-7 325T FPGA, 8-Slot CompactRIO Controller,NI-9205 C Series Voltage Input Module, 32-Ch, +/−10V, 250 kS/s/ch, 16-bit,NI 9264, 16-Ch voltage, +/−10V, 16-bit, 25kS/s/ch AO module,NI-9401 5 V/TTL, 8 Bidirectional Channels, 100 ns C Series Digital Module.

### 2.2. Control Strategy

The control strategy must command the actuator in order to achieve precise positioning, minimize the error, and deal with system uncertainties. To compensate for time delay and overcome the nonlinearities and disturbances mentioned in Section 2.1, this paper proposes a control strategy that combines an inverse piezoelectric model with a closed-loop control SMC-WDC. Inverse control adjusts the output while the SMC-WDC closed loop corrects possible errors or external disturbances taking into account the time-delay system behavior. In this section, both the inverse control and SMC-WDC control solutions are described. The proposed control scheme is shown in Figure 4.

#### 2.2.1. Inverse Control

Inverse control is based on an inverse model that represents mathematically the behavior of a system. To build a complete model of the system, several semi-empirical models were combined. The semi-empirical models were constructed based on the mathematical equations that reproduced the system behavior. Some empirical parameters were estimated to adjust the model. 

##### System Mathematical Model

The proposed system mathematical model took into account the two main nonlinearities mentioned in Section 2.1.2—hysteresis and dynamical behavior—which included internal damping and friction forces. As shown in Figure 2, the positioning system can be modelled in three elements. If the PEA stiffness is assumed to be much higher than the PEA amplifier mechanism stiffness and the positioning mechanism stiffness, the simplified system can be modeled as a second-order model with a single element, as shown in Figure 5.

In this case, the equation that correlates the PM displacement ($x_{PM}$) with the PEA displacement ($x_{PEA}$) can be written as follows:(1)$$b_{PAM}\left({\dot{x}}_{PEA}\left(t\right)-{\dot{x}}_{PM}\left(t\right)\right)+k_{PAM}\left(x_{PEA}\left(t\right)-x_{PM}\left(t\right)\right)=(m_{PAM}+m_{PM}){\ddot{x}}_{PM}\left(t\right)+b_{PM}{\dot{x}}_{PM}\left(t\right)+k_{PM}x_{PM}\left(t\right).$$

Additionally, the PEA displacement can be modeled as follows: (2)$$x_{PEA}=u\left(t\right)d-h\left(t\right),$$
where *u* is the PEA input voltage, *d* is the piezoelectric constant, and *h* is the hysteresis correction. In the model proposed in this article, the piezoelectric constant *d* is represented for a third-order equation *D*, as in [25]. This term tries to achieve a more precise relationship between the displacement and the applied voltage. The voltage–displacement relationship has been modeled by the following equation:(3)$$u\left(t\right)d=d_{1}u\left(t\right)+d_{2}u{\left(t\right)}^{2}+d_{3}u{\left(t\right)}^{3}.$$

In Equation (1), the PM and PAM masses as well as spring and damp constants can be grouped as element equivalent (EQ) constants as follows:(4)$$m_{EQ}=m_{PAM}+m_{PM},$$
(5)$$b_{EQ}=b_{PAM}+b_{PM},$$
(6)$$k_{EQ}=k_{PAM}+k_{PM}.$$

Thus, the resulting model is the following:(7)$$m_{EQ}{\ddot{x}}_{PM}\left(t\right)+b_{EQ}{\dot{x}}_{PM}\left(t\right)+k_{EQ}x_{PM}\left(t\right)=b_{PAM}\left(\dot{u}\left(t\right)D-\dot{h}\left(t\right)\right)+k_{PAM}\left(u\left(t\right)D-h\left(t\right)\right).$$

The Bouc–Wen model is a simple frequency dependency model, which combines hysteresis and dynamic vibrations [36]. The hysteresis is modeled by means of three parameters (*α*, *β*, and *γ*) as follows:(8)$$\dot{h}\left(t\right)=\alpha d\dot{u}\left(t\right)-\beta \left|\dot{u}\left(t\right)\right|h\left(t\right){\left|h\left(t\right)\right|}^{n-1}-\gamma \dot{u}\left(t\right){\left|h\left(t\right)\right|}^{n},$$
where *u* is the applied voltage, *h* is the variable that describes the hysteresis, *d* is the effective coefficient of the piezoelectric, and *α*, *β*, and *γ* are the parameters that adjust the shape of the hysteresis loop. The linear constant *d* is maintained to preserve the straightforwardness of the correction by hysteresis. For an elastic structure and material, n = 1, Equation (8) then becomes the following:(9)$$\dot{h}\left(t\right)=\alpha d\dot{u}\left(t\right)-\beta \left|\dot{u}\left(t\right)\right|h\left(t\right)-\gamma \dot{u}\left(t\right)\left|h\left(t\right)\right|.$$

##### Parameter Estimation

The model parameters were estimated using empirical data from open-loop control measurements. The voltage input signal was a variable frequency sine (from 1 Hz to 10 Hz) and constant amplitude (9.2 volts). The voltage was amplified 7.5 volts/volt by the voltage amplifier.

First, the *d* and *D* coefficient values were estimated with the Excel trendline function: *d* = 40.45 and *D: d1* = 53.82, *d2* = −6.9361, *d3* = 0.8461.

The values of the constants *m_EQ_, b_EQ_, b_PAM_, k_EQ_, k_PAM_, α, β*, and *γ* were estimated using the MATLAB R2017b parameter estimation function based on the least squares method.

As shown in Table 1, the estimated *b_PAM_* constant is minimum with respect to *b_EQ_*. Consequently, this parameter can be neglected and Equation (7) can be simplified as follows:(10)$$m_{EQ}{\ddot{x}}_{PM}\left(t\right)+b_{EQ}{\dot{x}}_{PM}\left(t\right)+k_{EQ}x_{PM}\left(t\right)=k_{PAM}\left(u\left(t\right)D-h\left(t\right)\right).$$

The comparison between the experimental measurements and the simulated results (using the above estimated constants) is shown in Figure 6. A slightly deviation is observed, which can be corrected with the introduction of an appropriate closed-loop control scheme.

#### 2.2.2. Sliding Mode Control with Dynamic Correction

Open-loop control is a solution that offers good results, but the error caused by uncertainties and external perturbations is not corrected, so using closed-loop controls could notably improve the accuracy. The authors propose in this paper the use of a control based on the SMC technique, which is a robust strategy against model imperfections and uncertainties.

##### Sliding Mode Control

SMC is a nonlinear control approach that drives the state trajectory of the system onto a specified sliding surface and maintains the trajectory on that surface for the subsequent time. However, in conventional SMC design, a priori knowledge of the bounds on system uncertainties must be acquired [25]. Perturbation estimation strategies have already been studied in the literature [10,25]. One common approach to estimating perturbation *p_est_* is as follows:(11)$$p_{est}\left(t\right)=m_{EQ}{\ddot{x}}_{PM}\left(t\right)+b_{EQ}{\dot{x}}_{PM}\left(t\right)+k_{EQ}x_{PM}\left(t\right)-k_{PAM}\left(u\left(t-T\right)D-h\left(t-T\right)\right).$$

Then, the system model becomes the following:(12)$$m_{EQ}{\ddot{x}}_{PM}\left(t\right)+b_{EQ}{\dot{x}}_{PM}\left(t\right)+k_{EQ}x_{PM}\left(t\right)=k_{PAM}\left(u\left(t-T\right)D-h\left(t-T\right)\right)+p_{est}\left(t\right)+\tilde{p}\left(t\right),$$
where $\tilde{p}\left(t\right)=p\left(t\right)-p_{est}\left(t\right)$ represents the error between the system’s real and estimated perturbations. To design the SMC controller, the position error is defined as follows:(13)$$e\left(t\right)=x\left(t\right)-x_{d}\left(t\right),$$
where *x_d_* represents the position reference and t denotes the time variable. In the rest of this section, the time indices have been omitted for the sake of brevity. As the dynamic system of the PEA is a second-order system, a first-order PD sliding surface was selected:(14)$$s=\dot{e}+\lambda e,$$
where λ (λ > 0) is a design parameter.

The control law for system (12), with sliding surface (14) and the position error given by (13) which satisfies $\underset{t\to 0}{\lim}\;e\left(t\right)=0$, is as follows, as proven in [10]:(15)$$u=\frac{m}{kd}\left(\frac{b}{m}-\lambda \right){\dot{x}}_{PM}+\frac{1}{d}\left(x_{PM}+h\right)-\frac{1}{kd}p_{est}+\frac{m}{kd}\left({\ddot{x}}_{d}+\lambda {\dot{x}}_{d}\right)-\eta \;sgn\left(s\right),$$
where *η* is a positive switching gain and *sgn(s)* represents the signum function, as defined in the following:(16)$$sgn\left(s\right)=\left\{\begin{array}{c} -1,\;\mathrm{for}\;s<0\\ 0,\;\mathrm{for}\;s=0\\ 1,\;\mathrm{for}\;s>0 \end{array}\right..$$

Due to the discontinuity of the sign function, the control input may have chattering. To reduce this phenomenon, the boundary layer technique was used by replacing the signum function by both saturation and hyperbolic functions. Additionally, as described in [21,25], in order to design an improved SMC controller, a proportional–integral–derivative PID-type sliding surface can be defined. 

##### SMC with Dynamical Correction

In the case of systems with no time delay in the response, SMC proved to be an effective control strategy [10,25]; however, when there is a delay in the response, the control does not perform adequately. As seen in the introduction, there are several strategies to improve control in time-delay systems [29,30,31,32,33,34].

The positioning reference and the actuator are situated in different locations, therefore controlling the PEA based directly on the PM error (*e_PM_*) is not the best approach to overcome delay issues. At a certain moment when the system is still responding, the actuation reference may be in the correct position but not the reference position; in which case the closed loop would command an incorrect controlling signal. Traditional SMC schemes do not take into account mechanical correlation, so the actuation response leads to chattering and an unstable response of the system.

To correct the lack of coordination between the actuator and the system response, a dynamical correction on the sliding surface *s* is proposed in this paper. The goal was to command the actuator based on an equivalent positioning error in the PEA (*e_PEA_*) that takes into account the response of the whole system, instead of on a downstream PM error (*e_PM_*). The error being the difference between the desired setpoint and the actual position, the error in the positioning system *e_PM_* and the error in the PEA *e_PEA_* can be written as follows:(17)$$\begin{array}{c} e_{PM}=x_{dPM}-x_{PM},\\ e_{PEA}=x_{dPEA}-x_{PEA}. \end{array}$$

If we assume a positioning error on the reference PM of *e_PM_*, the error can be correlated with the equivalent PEA error *e_PEA_* using the mechanical system equation, which is presented in Equation (18). Thus, the SMC input error is replaced with the equivalent PEA estimated error that needs to be corrected, as follows:(18)$$m_{EQ}{\ddot{e}}_{PM}+b_{EQ}{\dot{e}}_{PM}+k_{EQ}e_{PM}=k_{PAM}e_{PEA}.$$

Hence the proposed control law (15) using a hyperbolic function and the proposed dynamical correction gives the following:(19)$$u=\frac{m}{kd}\left(\frac{b}{m}-\lambda \right){\dot{x}}_{PM}+\frac{1}{d}\left(x_{PM}+h\right)-\frac{1}{kd}p_{est}+\frac{m}{kd}\left({\ddot{x}}_{d}+\lambda {\dot{x}}_{d}\right)-\eta \;\tanh\left({\dot{e}}_{PM}+\lambda e_{PM}\right).$$

## 3. Results

In this section, the performance of the proposed SMC-WDC control strategy is analyzed. The control is evaluated by observing the positioning and the error, given a defined reference positioning signal. For each analysis, graphs were used to analyze the system behavior. The SMC-WDC positioning performance was compared with different control strategies, namely PI, SMC with saturation (SMC-Sat), and SMC with hyperbolic function (SMC-Hyp), all of them in combination with inverse open-loop control. The reference signals used for these experiments were a ramp signal and a sinusoidal signal. Below, the control configuration is first described and then the experimental results with the ramp and sinusoidal references are analyzed.

### 3.1. Control Configuration

To precisely control the micropositioning system, a custom program was created in LabVIEW 2017. The monitoring and control loops, which were executed in the field-programmable gate array (FPGA) target, ran at 50 kHz and 5 kHz, respectively. Real-time (RT) loops were responsible for transmitting the commands to the FPGA and saving the data received from the FPGA into a file. In the RT, the voltage output, confocal laser position signal, and setpoint data were registered at 500 Hz. The RT was also responsible for monitoring the status of the FPGA. Finally, the user interface control was executed on a desk computer.

The control gains were adjusted experimentally in the four control strategies. Several trials were carried out to tune the control gains adequately and obtain the optimal response of the controller. Low gain values produced high error and slow system response. On the other hand, it was observed that using proportional control gains that were too high led to chattering and an unstable response of the system. Additionally, the use of high integral control gains led to an additional delay response of the controller, especially on the SMC-WDC; we found that it produced the countereffect inn the dynamical correction. For this reason, the integral gain in the proposed SMC-WDC scheme was minimized to 0.0001, which produced a minimal filtering effect action and rejected high frequency confocal laser sensor noise. The voltage signal was limited in all controllers to the recommended range for the actuator, that is, 0 to 75 V. The value η determines the SMC switching gain. A 75 V range was selected to ensure a better stabilization against any disturbance and nonlinearity, although this could cause high activity control. The boundary layer δ for saturation and hyperbolic limits ensures the response of the SMC from a minimum error, which was set to 1 µm. The selected control parameters values are listed in Table 2.

### 3.2. Ramp Signal Tracking Test

The purpose of the test was to evaluate the control strategies during sudden changes of speed and determine the stabilization time with a ramp reference signal. This signal is divided into three steps: (1) a 200 µm/second ramp from 0 µm to 200 µm, (2) a 1 s stabilization constant reference of 200 µm, and finally (3) a 200 µm/second ramp back to the origin 0 µm. As is shown in Figure 7a, all the control responses are relatively accurate and all the control strategies reproduce the ramp reference shape. 

Detailed control performance can be observed in the tracking error graph in Figure 7b. Independently of the control strategy, error is always comprised between −7 µm and 22 µm. The response of the inverse open loop with SMC with saturation is slow, resulting in a peak error of 22 µm. However, the stabilization provided by the integral gain reduces the error to 1 µm. The SMC with hyperbolic function has a similar performance; however, the control response is much faster and the maximum peak error is 8 µm. Compared with traditional PI, the error is minor in SMC-Hyp control. SMC-WDC presents the fastest response with the tracking error occurring between −5 µm and 3 µm. Stabilization is achieved after 1 s and the error is reduced to less than 2 µm. A higher integral gain would reduce the permanent error, although it would decrease the system response significantly.

The control response can be examined on the detailed graph presented in Figure 8. The reference ramp starts at 0.02 s. The SMC-WDC control provides the best response since the mechanism displacement increases at 0.03 s, which is a 0.01 s delay. PI and SMC-Hyperbolic control react after 0.03 s and 0.046 s, respectively. The worst response is given by the SMC with saturation control, with a time delay of 0.112 s.

### 3.3. Sinusoidal Signal Tracking Test

The response of the control against variable low frequency signal was evaluated using a sinusoidal signal as setpoint. The sinusoidal reference signal was configured with a phase delay of –π/2 and an offset equal to the amplitude. There was no steep change in the reference signal. The amplitude was 100 µm and the frequency was 1 Hz constant. The control responses were accurate and all the control strategies reproduced the sinusoidal reference shape without significant deviations, as shown in Figure 9a.

Detailed control performance can be observed in the tracking error graph in Figure 9b, where the error occurred from −7 µm and 7 µm. Inverse open loop with PI, SMC with saturation, and SMC with hyperbolic function had a similar response, that is, the maximum error appears while the speed is maximum and, in areas where the position does not have a great variation, the integral gain reduces the error. The response of SMC-WDC control is again the best, where we observed a better response when the reference speed increased. When the reference speed was minimum, the SMC-WDC with low integral gain did not reduce the error so fast.

In Figure 10 the responses during the first 0.2 s can be analyzed. The SMC-WDC shows a response similar to the rest of the controls. In the sinusoidal signal reference, no big change occurs in the speed reference, therefore the traditional control strategies do not present significant deviations. As the reference speed increases, the SMC-WDC control approximation to the reference is fastest. 

## 4. Conclusions

Time-delay micropositioning systems present a great challenge when trying to achieve accurate positioning because the control strategy has to cope with a complex mechanism and its nonlinearities to achieve a successful positioning. Traditional feedback controls have a tendency to overcompensate and high controller gains must be rejected. The integral gain reduces the permanent error, although it increases the system delay and causes inaccurate positioning while following dynamical references. Since inappropriate control strategies may lead to chattering and positioning deviations, the control strategy must take into account the response of the system in order to effectively compensate for uncertainties and disturbances.

In this paper, a mixed control strategy for time-delay micropositioning systems was presented, which combined inverse open-loop control and sliding mode control with dynamical correction. The SMC-WDC, which was based on an SMC with hyperbolic function, employed an estimated equivalent error *e_PEA_* and prevented the controller from using incorrect input data. The SMC-WDC control strategy took into account the necessary corrections on the PEA that later become effective. The SMC-WDC results were especially effective when abrupt setpoint variations were required. 

The micropositioning mechanism shown in this paper was used to evaluate different control strategies. Combining inverse open-loop control with traditional PI and SMC closed-loop controls made reasonable results possible, that is, ramp and sinusoidal tests showed that the error occurred between −7 µm and 7 µm, and the response time was 0.03 s against sudden changes of speed reference. Nevertheless, the proposed SMC-WDC control strategy was proven to offer the best results, that is, among the tests the error occurred between −5 µm and 5 µm, and the response time was 0.01 s against sudden changes of speed reference. The SMC-WDC, compared to PI and SMCs, provides a remarkably fast response time and a minor positioning error. 

## Figures and Tables

**Figure 1 materials-13-00132-f001:**
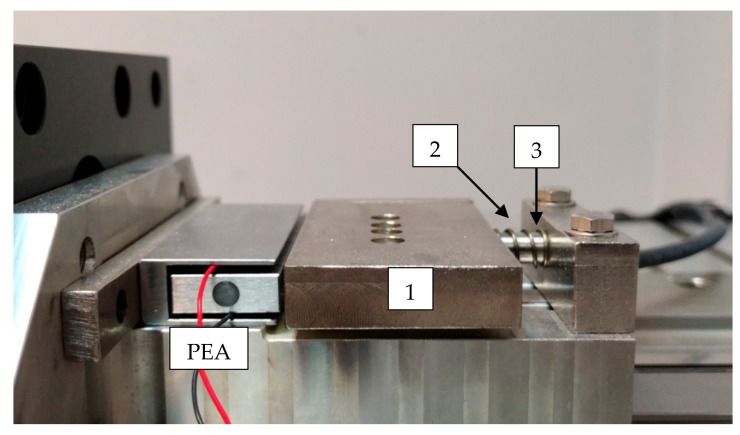
Positioning system setup: (1) main body, (2) spring holder, and (3) pin fixture.

**Figure 2 materials-13-00132-f002:**
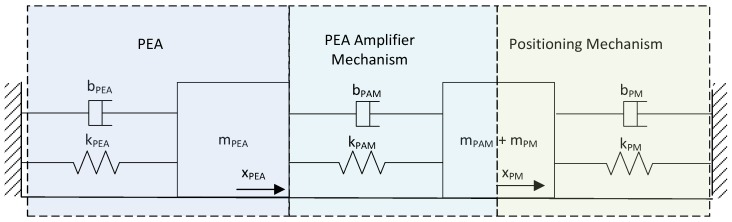
Positioning system mechanical model.

**Figure 3 materials-13-00132-f003:**
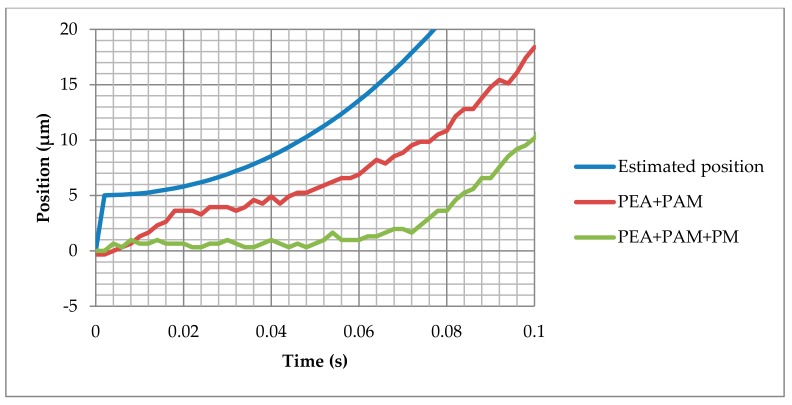
Open-loop system response comparison: Ideal system (Estimated position = Input Voltage × Displacement Constant), amplified piezoelectric actuator (piezoelectric actuator (PEA) + PEA amplifier mechanism (PAM)), and complete positioning system (PEA + PAM + positioning mechanism (PM)).

**Figure 4 materials-13-00132-f004:**
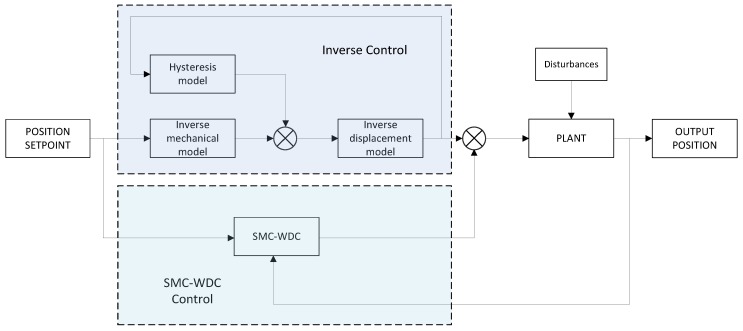
Combined control strategy.

**Figure 5 materials-13-00132-f005:**
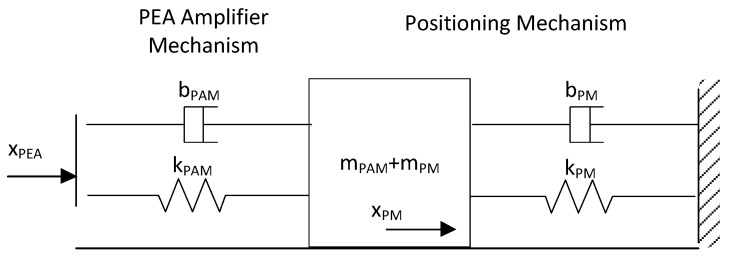
Simplified mechanical model for the positioning system.

**Figure 6 materials-13-00132-f006:**
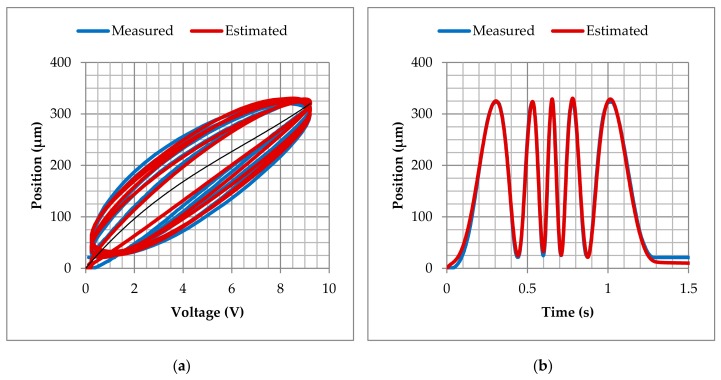
PEA system model in comparison with experimental data: (**a**) position versus voltage and (**b**) position versus time.

**Figure 7 materials-13-00132-f007:**
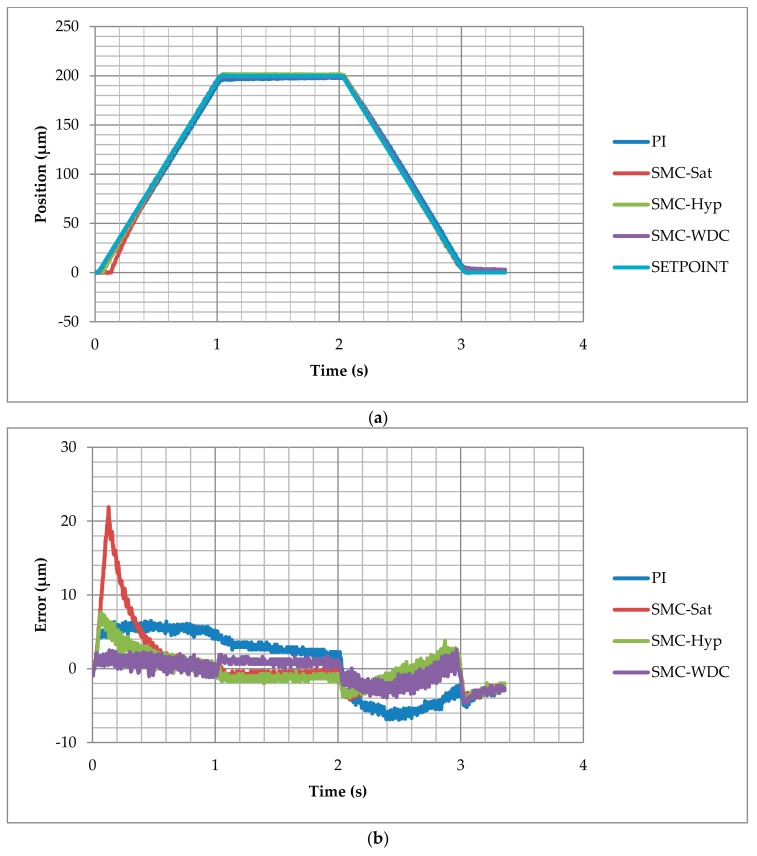
Ramp test results of integral proportional control (PI), sliding mode control (SMC) with saturation (SMC-Sat), SMC with hyperbolic function (SMC-Hyp), and inverse dynamic estimation SMC (SMC-WDC) tracking the setpoint (SETPOINT) along the time: (**a**) position and (**b**) tracking error.

**Figure 8 materials-13-00132-f008:**
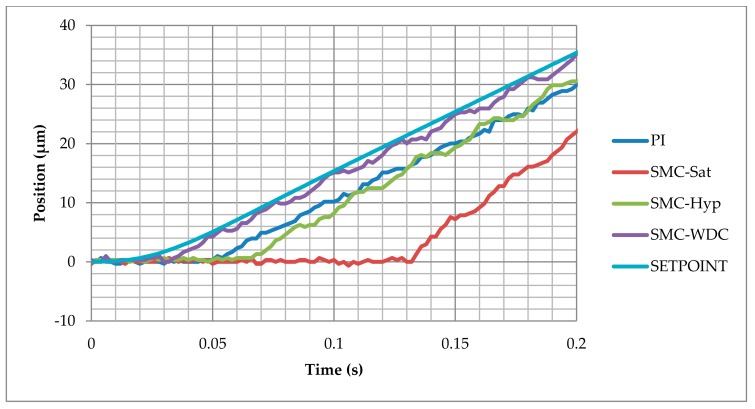
Response delay comparison between the different control strategies in the ramp test.

**Figure 9 materials-13-00132-f009:**
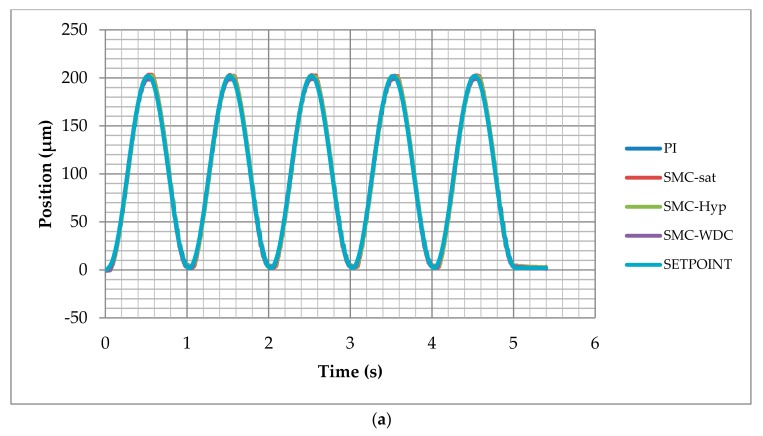
Sinusoidal test results of PI, SMC with saturation (SMC-Sat), SMC with hyperbolic function (SMC-Hyp), and inverse dynamic estimation SMC (SMC-WDC) tracking the setpoint (SETPOINT) along the time: (**a**) position and (**b**) tracking error.

**Figure 10 materials-13-00132-f010:**
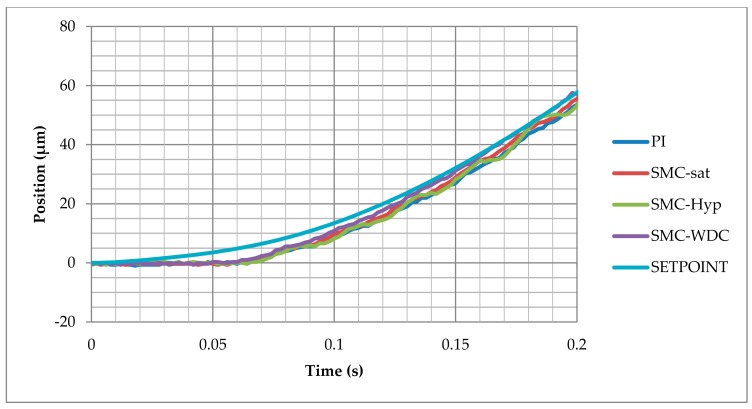
Response delay comparison between the different control strategies in the sinusoidal test.

**Table 1 materials-13-00132-t001:** Constant parameter estimated values.

Constant	Value	Units
*m_EQ_*	4.1 × 10^−1^	kg
*b_EQ_*	8.01 × 10^2^	Ns/m
*b_PAM_*	1.50 × 10^−12^	Ns/m
*k_EQ_*	5.84 × 10^3^	N/m
*k_PAM_*	5.23 × 10^3^	N/m
*α*	4.5 × 10^−1^	-
*β*	4 × 10^−1^	-
*γ*	7.29 × 10^−17^	-

**Table 2 materials-13-00132-t002:** Control parameter values.

Control	P	I	δ	η
PI	70,000	10	-	-
SMC-Sat	6000	10	1 µm	75 V
SMC-Hyp	6000	5	1 µm	75 V
SMC-WDC	5000	0.0001	1 µm	75 V

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
