# Peer review of "Sliding Mode Control with Dynamical Correction for Time-Delay Piezoelectric Actuator Systems"

_materials, 2019, doi:10.3390/ma13010132_

Round 1

Reviewer 1 Report

Comment 1:

The problem studied in this paper is quite a matured topic. The authors need to prove the novelty of their work.

Comment 2:

In the first paragraph, use of piezoelectric actuators in applications other than positioning, e.g., the piezoelectric friction and inertia actuators, see the paper “Development of a two-degree-of-freedom piezoelectric rotary-linear actuator with high driving force and unlimited linear movement”, where a two degrees of freedom motor is developed. A comprehensive literature review on the piezoelectric friction-inertia actuators can be found in the following paper “Piezoelectric friction–inertia actuator—a critical review and future perspective”. https://link.springer.com/article/10.1007/s00170-011-3827-z

Comment 3:

Quoted: “Then, a mixed control strategy, combining inverse open loop control and SMC-WDC, has been developed.”

Comment: it is not clear about “inverse open loop control”. In the sentences above, you only mentioned SMC-WDC. Does “inverse open loop control” refers to WDC?

Comment 4:

Time delay is a phenomenon, and there is a cause of it. The paper does not seem to clearly describe the cause. In fact, in piezoelectric material, the time delay can be caused by hysteresis and creep. In the piezoelectric friction-inertia actuator, time delay is also caused by friction. In this paper, friction does not seem to be paid attention. In micro-positioning system, friction phenomenon can be very complex, see the literature “experimental comparison of five friction models. ……” https://www.mech-sci.net/6/15/2015/

Author Response

Response to the reviewer comments

First of all, the authors would express their sincere gratitude to the Editors and the Reviewers who gave us many constructive comments and valuable suggestions in order to improve this paper. The authors have revised the paper according to the reviewers’ comments and the changes made in the paper have been done using the track changes in the word document. The responses to the reviewer comments can be found below their respective comments.

Comment 1:

The problem studied in this paper is quite a matured topic. The authors need to prove the novelty of their work.

Response 1:

The text in the abstract and Introduction has been reviewed to enhance the scientific advances presented in this paper: a SMC with dynamical correction to allow an increased performance against other control strategies. The models results have been tested experimentally in a commercial PEA. Check “Track Changes” on the word document.

Comment 2:

In the first paragraph, use of piezoelectric actuators in applications other than positioning, e.g., the piezoelectric friction and inertia actuators, see the paper “Development of a two-degree-of-freedom piezoelectric rotary-linear actuator with high driving force and unlimited linear movement”, where a two degrees of freedom motor is developed. A comprehensive literature review on the piezoelectric friction-inertia actuators can be found in the following paper “Piezoelectric friction–inertia actuator—a critical review and future perspective”. https://link.springer.com/article/10.1007/s00170-011-3827-z

Response 2:

The additional literature have been added to the introduction to include additional information about PEA applications and the analysis and modelling of PFIA mechanisms.

Comment 3:

Quoted: “Then, a mixed control strategy, combining inverse open loop control and SMC-WDC, has been developed.”

Comment: it is not clear about “inverse open loop control”. In the sentences above, you only mentioned SMC-WDC. Does “inverse open loop control” refers to WDC?

Response 3:

It was a failure, the SMC-WDC uses the dynamical model to adapt the inputs, not the inverse. The abstract has been corrected and tried to clarify SMC-WDC: in this paper the SMC-WDC has been tested in combination with open loop control. Additionally, the explanation on section “2.2.1. Sliding Mode Control with Dynamic Correction” now includes the proposed SMC-WDC control law to show the final control scheme.

Comment 4:

Time delay is a phenomenon, and there is a cause of it. The paper does not seem to clearly describe the cause. In fact, in piezoelectric material, the time delay can be caused by hysteresis and creep. In the piezoelectric friction-inertia actuator, time delay is also caused by friction. In this paper, friction does not seem to be paid attention. In micro-positioning system, friction phenomenon can be very complex, see the literature “experimental comparison of five friction models. ……” https://www.mech-sci.net/6/15/2015/

Response 4:

The analysis of the system response in section 2.1.2. Positioning Mechanism has been extended. The PM additional masses, springs and frictions cause the delay. It has been included on the Introduction a literature reference which found a time delay/ time lag on amplified PEA with additional clampings.

Reviewer 2 Report

This paper provides a method for sliding mode Control for time delay piezoelectric actuator systems. The analytical model and the corresponding equations are well explained and the results and conclusions are clearly presented. It is recommended to improve the style of the presentation of the results and to use white as background color instead of green.

The base line in Figure 5 is irritating, since it seems to couple the Input xPEA with xPM. It is recommended to leave it out.

The language sometimes lacks good style and contains some spelling or grammar mistakes, some of them are stated below together with a few hints for improvement:

Abstract: hz => Change to Hz
page 2, line 71: In case the system do => In case the system does
page 2, line 74: >10Hz => > 10 Hz (same error in other places)
page 2, line 79: an sliding mode control => a sliding mode control

2.1, line 102: consists on => consists of
2.1.4, line 145: is that measures => is that it measures
2.2.1, line 168: several models has been => several models have been
page 4, 199 hz => Hz

page 6, line 248: leads on => leads to (?)
page 7, line 272 and 284 and 288: leave blank between number and unit

page 7, line 295: a final a => finally a

page 9, line 314 check Grammar of last sentence.

page 9, line 320: hertz => Hz

A few sentences have awkward grammar. Please check. E.g. page 8, line 304

Author Response

Response to the reviewer comments

First of all, the authors would express their sincere gratitude to the Editors and the Reviewers who gave us many constructive comments and valuable suggestions in order to improve this paper. The authors have revised the paper according to the reviewers’ comments and the changes made in the paper have been done using the track changes in the word document. The responses to the reviewer comments can be found below their respective comments.

Comment 1:

This paper provides a method for sliding mode Control for time delay piezoelectric actuator systems. The analytical model and the corresponding equations are well explained and the results and conclusions are clearly presented. It is recommended to improve the style of the presentation of the results and to use white as background color instead of green.

Response 1:

In Section 3 Results the graphs are with white as background with gridlines. In figure 2 and figure 4 there have been used light colors to distinguish the different parts of the diagrams.

Comment 2:

The base line in Figure 5 is irritating, since it seems to couple the Input xPEA with xPM. It is recommended to leave it out.

Response 2:

The line have been reduced to clarify the simplified mechanism. It has not been erased completely since the authors believe it is useful to explain the mechanical relationship between the PM and PEA displacements.

Comment 3:

The language sometimes lacks good style and contains some spelling or grammar mistakes, some of them are stated below together with a few hints for improvement:

Abstract: hz => Change to Hz

page 2, line 71: In case the system do => In case the system does

page 2, line 74: >10Hz => > 10 Hz (same error in other places)

page 2, line 79: an sliding mode control => a sliding mode control

2.1, line 102: consists on => consists of

2.1.4, line 145: is that measures => is that it measures

2.2.1, line 168: several models has been => several models have been

page 4, 199 hz => Hz

page 6, line 248: leads on => leads to (?)

page 7, line 272 and 284 and 288: leave blank between number and unit

page 7, line 295: a final a => finally a

page 9, line 314 check Grammar of last sentence.

page 9, line 320: hertz => Hz

A few sentences have awkward grammar. Please check. E.g. page 8, line 304

Response 3:

Taking into account the reviewer comments, the paper has been revised carefully and minor syntax errors has been corrected.

Round 2

Reviewer 1 Report

i am happy with the revision.